# Frequency modulation on magnons in synthetic dimensions

Meng Xu [1], Chensong Hua[2], Yan Chen[1,3] ✉ & Weichao Yu [2,3,4] ✉

Magnons are promising candidates for next-generation computing architectures, offering the ability to manipulate their amplitude and phase for information encoding. However, the frequency degree of freedom remains largely unexploited due to the complexity of nonlinear process. In this work, we introduce the concept of synthetic frequency dimension into magnonics, treating the eigenfrequency of inherent modes as an additional degree of freedom. This approach enables the effective description of the temporal evolution of a magnon state using an effective tight-binding model, analogous to a charged particle hopping in a modulated lattice. A magnonic ring resonator is investigated as an example, and several intriguing phenomena are predicted, including Bloch oscillations and a leverage effect during unidirectional frequency shifts, all of which are verified through micromagnetic simulations. Notably, our strategy operates in the linear spin-wave regime, excluding the involvement of multi-magnon scattering and high-power generation. This work expands the toolkit for designing magnonic devices based on frequency modulation and paves the way for a new paradigm called magnonics in synthetic dimensions.

Magnons or spin waves are elementary excitations in magnetic systems free from Joule heating, which are promising information carriers for building magnonic circuits[1,2] with potential applications for classical[3], neuromorphic[4] and other unconventional computing architectures[5], etc. During past few decades, efforts have been made to explore possible ways to encode information into intrinsic degrees of freedom of magnons. Similar to other wave counterparts such as acoustic waves and electromagnetic waves, there are mainly four routes to encode information in spin waves: (i) Amplitude, where binary information 1(0) is encoded into spin waves with high (low) amplitude and by manipulation of which one can design unidirectional devices and logic gates[6,7]. (ii) Phase, where the concept of wave interference is harnessed and devices like Mach-Zehnder-type spin wave interferometer[8] and majority gates[9] are proposed. (iii) polarization, which is a degree of freedom possessed by antiferromagnets and ferrimagnets with two opposite sublattices[10,11]. (iv) Frequency, where information is encoded into spectral components of time-varying

signals with typical application of magnonic frequency combs[12–16]. Despite the widespread use of frequency modulation in radio communication benefiting from its great bandwidth efficiency and less susceptibility to interference, there have been few magnonic devices utilizing this concept due to lack of strategies to perform frequency modulation on magnons efficiently. Recently, the development of nonlinear magnonics[17] offers possible ways to manipulate magnonic state in frequency domain. One typical strategy is to realize spectral shift when a magnetic system is driven into nonlinear regime by high-power inputs[18–21]. Another strategy is to trigger the nonlinear scattering process between magnon modes and other modes, such as breathing mode of magnetic skyrmion[14,22], cavity photons[23,24] and a pump-induced magnon mode[13,25]. Unfortunately, the involvement of nonlinear process requires that the amplitude of spin waves must be larger than a certain threshold, leading to instability and chaotic dynamics[17], against the intention of energy efficiency and feasible controllability.

[1]Department of Physics, Fudan University, Shanghai 200433, China. [2]Institute for Nanoelectronic Devices and Quantum Computing, Fudan University, Shanghai 200433, China. [3]State Key Laboratory of Surface Physics, Fudan University, Shanghai 200433, China. [4]Zhangjiang Fudan International Innovation Center, Fudan University, Shanghai 201210, China. ✉e-mail: yanchen99@fudan.edu.cn; wcyu@fudan.edu.cn

In this work, we apply the concept of *synthetic dimension* to perform frequency modulation on magnons in linear regime. Instead of shifting the spectrum itself, the proposed strategy aims to redistribute magnon occupation on the spectrum by using a temporally periodic driving field, in accordance with Floquet engineering[26–29]. The concept of synthetic frequency dimension was first proposed by Yuan et al.[30–34], which has been successfully applied in photonics[35–39] and atomic trap systems[40]. The key spirit of synthetic dimension is to treat an intrinsic degree of freedom, e.g., the frequency dimension, on an equal footing with spatial dimension, so that one can manipulate temporal dynamics of excitations such as photons and magnons in the perspective of particle transport in an one-dimensional lattice.

As depicted in Fig. 1a, we consider a two-dimensional magnonic ring resonator with central radius $R$ and width $w$. The magnetizations are uniformly oriented along $\hat{z}$ by an external field $H$. Such a ring resonator supports unequally spaced resonant modes due to the group velocity dispersion naturally preserved by spin waves, as shown in Fig. 1b. Denoting $n$ as mode number for resonant modes with eigenfrequencies $\omega_n$, each mode can be treated as a lattice site in the synthetic frequency dimension, so that the transition of magnonic states in a ring resonator is equivalent to hopping of magnons in a synthetic one-dimensional lattice, as illustrated in Fig. 1c. In order to trigger magnon hopping between synthetic lattices, a dynamical modulation is needed which is proposed here to be an alternating voltage applied on the yellow region in Fig. 1a, based on the concept of voltage-controlled magnetic anisotropy (VCMA)[41–43], thus inducing a local change of effective field.

## Results

### Effective tight-binding model

According to the micromagnetic theory[44], the dynamics of the unit magnetization $\mathbf{m}$ is governed by the Landau-Lifshitz-Gilbert (LLG) equation[44]

$$\dot{\mathbf{m}}(\mathbf{r}, t) = -\gamma \mathbf{m}(\mathbf{r}, t) \times \mathbf{H}_{\text{eff}} + \alpha \mathbf{m}(\mathbf{r}, t) \times \dot{\mathbf{m}}(\mathbf{r}, t), \quad (1)$$

with gyromagnetic ratio $\gamma$ and Gilbert damping coefficient $\alpha$. The effective field $\mathbf{H}_{\text{eff}} = A\nabla^2 \mathbf{m} + H\hat{z}$ includes exchange interaction with exchange coefficient $A$ and external field $H$. The demagnetizing field is not considered here for simplicity, which is valid for high-frequency

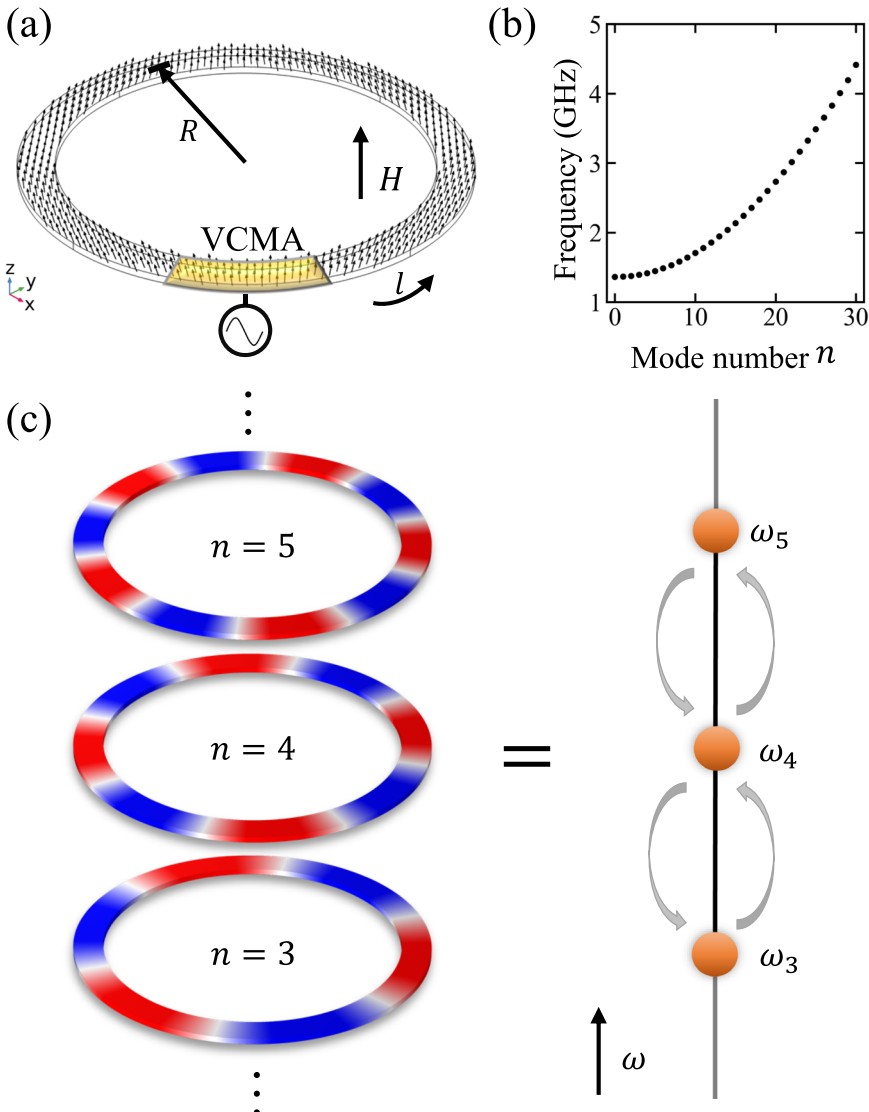

**Fig. 1 | Synthetic frequency dimension in a magnonic ring resonator.**
**a** Schematics for a magnonic ring resonator attached by a modulator with alternating voltages. **b** Resonant modes of the magnonic resonator in **a** with unequal frequency spacing. **c** Mode number can be treated as discrete lattices in synthetic frequency dimension so that the evolution of spin-wave excitation is equivalent to a particle hopping in the synthetic lattices.

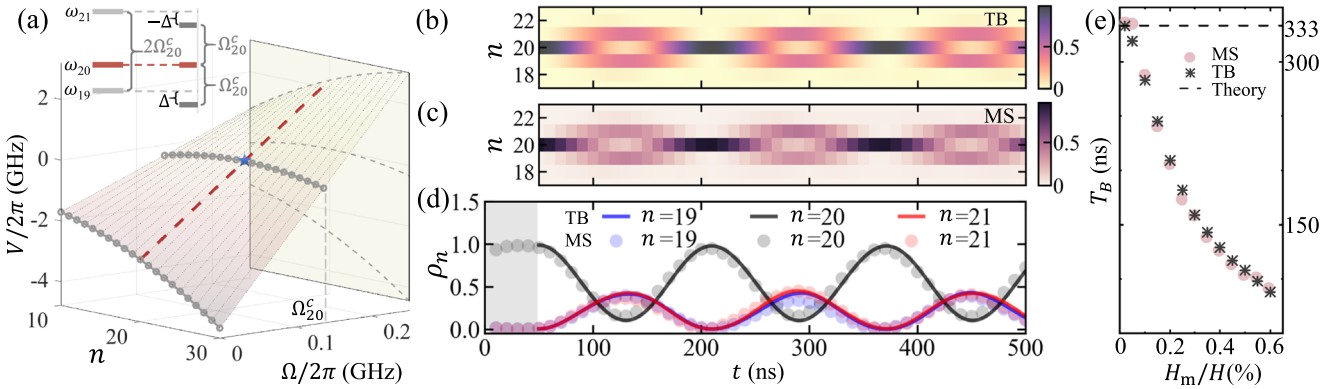

**Fig. 2 | Magnonic Bloch oscillation in synthetic frequency dimension. a** Effective onsite potential of magnons evolving on a synthetic frequency lattice under dynamical modulation. The inset illustrates the energy diagram for a magnon state initially placed at $\omega_{20}$ when Bloch oscillation is achieved with the critical driving frequency $\Omega = \Omega_{20}^c$. **b–d** Magnonic Bloch oscillation in synthetic frequency dimension, calculated by **b** tight-binding (TB) model and **c** micromagnetic simulation (MS), where a dynamical modulation with strength $H_m$ and angular frequency $\Omega = \Omega_{20}^c$ is continuously applied after $t = 50$ ns. The occupation of magnon state is characterized by probability density $\rho_n$ from TB and normalized spin-wave spectrum from MS. **e** Bloch period $T_B$ versus modulation strength $H_m$. The dashed line indicates theoretical Bloch period $T_B = 2\pi/|\Delta_{20}|$ for a synthetic lattice with equally spaced levels.

excitations where the exchange interaction dominates. Assuming spin waves are uniformly excited in radial direction and propagate tangentially along arc coordinate $l$, Eq. (1) can be linearized by decomposing the excitation into static and dynamical components as $\mathbf{m}(l, t) = \mathbf{m}_0 + \delta\mathbf{m}e^{i(\omega t - kl)}$, and one can obtain the linearized equation of motion for the right-handed precession mode in the absence of Gilbert damping

$$-\omega m_+ = -\gamma H m_+ - \gamma A k^2 m_+, \quad (2)$$

with $m_+ \equiv (\delta m_x + i\delta m_y)e^{i(\omega t - kl)}$. The dispersion of spin-wave eigenstates can be obtained from Eq. (2) that $\omega_n = \omega_0 + \gamma A k_n^2 = \omega_0 + \omega' n^2$ with $\omega_0 = \gamma H$, $\omega' = 4\pi^2\gamma A/L^2$, $k_n = (2\pi n)/L$ due to the periodic nature of the ring structure, where $L = 2\pi R$ is the mean circumference. It is seen in Fig. 1b that the magnonic ring resonator supports a set of modes with unequal frequency spacing, which is a distinct feature of magnons other than other excitations with linear dispersions such as photons[45].

Considering a dynamical modulation is applied with strength $H_m$, angular frequency $\Omega$ and initial phase $\phi$, Eq. (2) can be further rearranged as a Schrödinger-like equation

$$i\dot{m}_+ = \gamma A\nabla^2 m_+ - \gamma[H + H_m\cos(\Omega t + \phi)]m_+, \quad (3)$$

which governs the motion of wave function $m_+$ in a periodic time-varying potential.

In general, spin-wave excitation $m_+$ can be expressed as the superposition of all resonant modes with mode amplitude $C_n$, i.e.,

$$m_+(l, t) = \sum_n C_n(t)e^{i(\omega_n t - k_n l)}. \quad (4)$$

Substituting Eq. (4) into Eq. (3) by setting $\phi = 0$ and keeping leading-order terms with assumption that $\omega_n - \omega_{n-1} < \Omega < \omega_{n+1} - \omega_n$, one can obtain a coupled-mode equation (see detailed derivation in supplementary information)

$$i\dot{C}_n = g\left[e^{i(\omega_{n+1} - \omega_n - \Omega)t}C_{n+1} + e^{-i(\omega_n - \omega_{n-1} - \Omega)t}C_{n-1}\right], \quad (5)$$

with coupling strength $g = -\gamma H_m/2$ proportional to the modulation strength. We denote the magnon state of the ring resonator as $|\psi(t)\rangle = \sum_n C_n(t)a_n^\dagger|0\rangle$, with $a_n^\dagger$ ($a_n$) the creation (annihilation) operator applied on vaccum state $|0\rangle$. According to Eq. (5), $|\psi(t)\rangle$ evolves

following the Schrödinger equation $i\partial_t|\psi(t)\rangle = \mathcal{H}|\psi(t)\rangle$ with the effective Hamiltonian

$$\mathcal{H} = \sum_n g\left[a_n^\dagger a_{n+1}e^{i(\omega_{n+1} - \omega_n - \Omega)t} + a_{n+1}^\dagger a_n e^{-i(\omega_{n+1} - \omega_n - \Omega)t}\right]. \quad (6)$$

We perform gauge transformation on the creation (annihilation) operators $\tilde{a}_n^\dagger(t) = \exp[-i(\omega_n - n\Omega)t]a_n^\dagger(t)$, so that the Hamiltonian Eq. (6) can be gauge-transformed according to $\tilde{\mathcal{H}} = \mathcal{U}(t)^\dagger\mathcal{H}\mathcal{U}(t) - i\mathcal{U}^\dagger(t)d_t\mathcal{U}(t)$[46], which reads as

$$\tilde{\mathcal{H}} = g\sum_n\left(\tilde{a}_n^\dagger\tilde{a}_{n+1} + \tilde{a}_{n+1}^\dagger\tilde{a}_n\right) + \sum_n(n\dot{\Omega}t + n\Omega - \omega_n)\tilde{a}_n^\dagger\tilde{a}_n, \quad (7)$$

where we assume the frequency of modulation $\Omega(t)$ to be time-dependent without loss of generality. Equation (7) is the effective tight-binding Hamiltonian describing the dynamics of a particle in a synthetic lattice with a time-independent hopping rate $g$ and a site-dependent potential $V_n(t) = n\dot{\Omega}t + n\Omega - \omega_n$. Other forms of effective Hamiltonian with different physical interpretations can be obtained by choosing other gauges.

In the rest of the Letter, the effective tight-binding Hamiltonian Eq. (7) is solved by QuTiP[47] and the original LLG equation Eq. (1) is solved by Micromagnetics Module based on COMSOL Multiphysics[6,10,48,49]. Validations using MuMax3[50] with and without demagnetizing field can be found in supplementary information. The following micromagnetic parameters of Yittrium Iron Garnet (YIG) are considered[6]: exchange coefficient $A = 0.328 \times 10^{-10}$ A m, gyromagnetic ratio $\gamma = 2.21 \times 10^5$ Hz/(A/m), Gilbert damping $\alpha = 1.3 \times 10^{-4}$ and external field $H = 0.388 \times 10^5$ A/m which is equivalent to the intrinsic crystalline anisotropy. The central radius of the ring resonator $R = 575$ nm with width $w = 50$ nm. The dynamical modulation is applied to the sector region of the ring resonator with an angle of 20 degrees (see Methods for details of numerical methods).

## Bloch oscillation

Assuming that the frequency of dynamical modulation is temporally invariant, the effective onsite potential can be simplified as $V_n = n\Omega - \omega_n$ according to Eq. (7), which is shown in Fig. 2a, illustrating how the frequency-dependent energy landscape evolves under different modulation conditions. The gradient of onsite potential $\Delta_n = V_n - V_{n-1} = \Omega - (\omega_n - \omega_{n-1})$ plays as an effective electric field $E_{eff} \sim \partial V_n/\partial n$ applied on a charged particle[33]. When $\Omega/2\pi = 0$ GHz, the

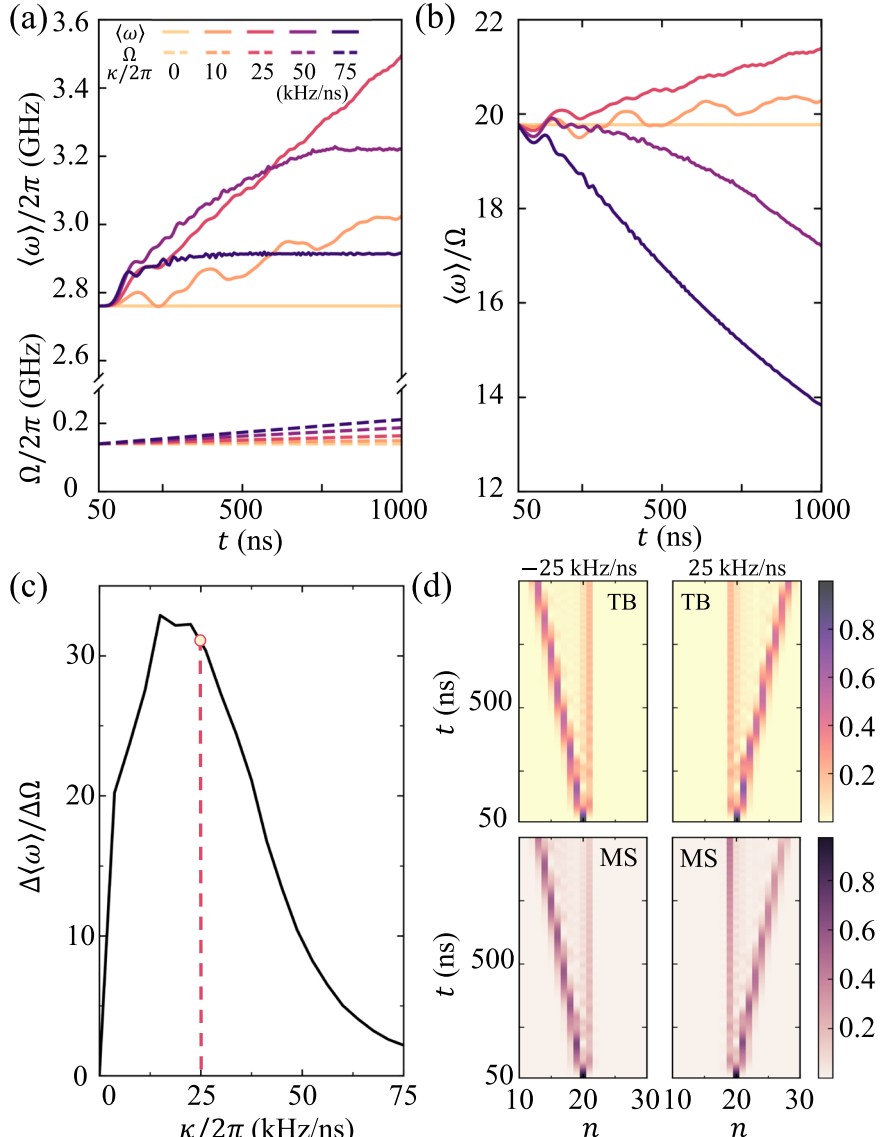

**Fig. 3 | Unidirectional frequency shift of magnon state. a** Temporal evolution for center of occupation $\langle\omega\rangle$ started from state $\omega_{20}$ under dynamical modulation with frequency linearly increasing with time $\Omega(t) = \Omega_0 + \kappa(t - t_0)$ which is applied after $t_0 = 50$ ns. **b** Frequency ratio $\langle\omega\rangle/\Omega$ versus time. **c** Leverage ratio $\Delta\langle\omega\rangle/\Delta\Omega$ versus changing rate of modulation $\kappa$. **d** Numerical results obtained from TB and MS methods for $\kappa/2\pi = \pm 25$ kHz/ns, demonstrating the symmetric frequency shift of magnon state in both directions.

spacing of onsite potential between neighbor sites forms an arithmetic progression due to the quadratic dispersion of exchange spin waves. In the presence of nonzero modulation strength $H_m$, it is expected that magnon occupation on synthetic lattices may experience complex evolution where analytical prediction is not always possible. However, regular evolution can be achieved at certain specific working point, e.g., *Bloch oscillation* of magnons in synthetic frequency dimension. The conventional realization of Bloch oscillation on charged particles requires a homogeneous electric field[51,52], or alternatively the presence of Wannier-Stark ladders with equidistant energies[53,54]. In our magnonic system, Wannier-Stack ladders can be approximately achieved when $|\Delta_n| = |\Delta_{n-1}|$, which gives the critical modulation frequency

$$\Omega_n^c = (\omega_{n+1} - \omega_{n-1})/2. \tag{8}$$

For the magnonic state initially placed at $\omega_{20}$, the critical modulation frequency is $\Omega_{20}^c/2\pi = 135$ MHz as labeled in Fig. 2a. The inset

of Fig. 2a specifically demonstrates how the critical modulation frequency $\Omega_{20}^c$ creates an energy configuration where the dressed state (under modulation) and the original magnon state at $\omega_{20}$ form equally-spaced energy levels, which is essential for achieving coherent Bloch oscillation. We further perform numerical calculation based on tight-binding (TB) model (Eq. (7)) and micromagnetic simulation (MS) (Eq. (1)). As shown in Fig. 2b–d, a magnonic state at $\omega_{20}$ is initially prepared and is characterized by probability density $\rho_n = |\langle n|\psi(t)\rangle|^2$ for tight-binding model and normalized spin-wave spectrum extracted from micromagnetic simulation (see supplementary information for numerical methods). A dynamical modulation with frequency $\Omega_{20}^c$ and strength $H_m = 0.3\%H$ is turned on at $t = 50$ ns. The magnon state initially localized at $\omega_{20}$ begins to spread across neighboring frequency modes, populating states from $\omega_{19}$ to $\omega_{21}$. This spreading and subsequent reconcentration occurs periodically with the Bloch period $T_B$. The quantitative agreement shown in Fig. 2d between two methods indicates the validity of the effective tight-binding model. The dynamical modulation provides the necessary energy compensation

through the coupling term $g$ in Eq. (7), enabling coherent transitions between adjacent modes with unequal frequency spacing while maintaining overall energy conservation through balanced energy exchange between the magnetic system and modulation device.

The dependence of Bloch period $T_B$ on the strength of dynamical modulation $H_m$ is further investigated. According to Eq. (7), the strength of modulation $H_m$ has no contribution to the onsite potential $V_n$ but plays a crucial role in the hopping rate $g$. With a larger $H_m$ (associated with larger $g$), the hopping range, i.e., the total range of frequency sites that a magnon state can sequentially access through nearest-neighbor hopping, will be extended in proportion to $2g/\Delta_n$[53]. Due to the unequal level spacing inherent in the magnonic system, when the magnon state undergoes transitions between neighboring sites, it experiences multiple hopping processes with different effective electric fields ($\cdots, \Delta_{18}, \Delta_{19}, \Delta_{21}, \Delta_{22} \cdots$, etc.). These effective fields are larger than $\Delta_{20}$ (the one at the initial state $\omega_{20}$). Consequently, when the modulation strength $H_m$ increases, the magnon state can hop more readily between sites experiencing these larger effective fields, resulting in a decreased Bloch oscillation period $T_B$. This behavior differs from conventional systems with equally spaced levels, where the Bloch period would remain constant regardless of hopping strength. The overall Bloch period is estimated using both TB and MS methods and the results are plotted in Fig. 2e, which agrees with the expectation. When $H_m$ approaches to zero, the Bloch period converges to the theoretical value $T = 2\pi/|\Delta_{20}| \simeq 333$ ns (dashed line in Fig. 2e) predicted for a conventional charged particle in a lattice with equally spaced potentials[53]. It is worth noting that when this hopping range becomes large enough, the assumption $\omega_n - \omega_{n-1} < \Omega < \omega_{n+1} - \omega_n$ no longer holds for all involved modes, and additional hopping processes beyond nearest-neighbor transitions become possible. However, these long-range hoppings would require different driving frequencies than those that enable nearest-neighbor transitions. Long-range hopping is inherently accounted for in the micromagnetic simulation but is not captured by the current model described in Eq. (7) (see supplementary information for discussion on long-range hopping process).

**Leverage effect for unidirectional frequency shift**

Other than periodic evolution such as Bloch oscillation, unidirectional shift of magnon state can be realized if the dynamical modulation is time-dependent. According to Fig. 2a, the effective onsite potential in the absence of dynamical modulation follows an arithmetic sequence, hence we propose that unidirectional frequency shift can be achieved by a dynamical modulation whose frequency increases or decreases linearly with time, i.e., $\Omega(t) = \Omega_0 + \kappa t$ with the initial driving frequency $\Omega_0$ and the changing rate $\kappa$. Figure 3a shows the unidirectional evolution of magnon state starting from $\omega_{20}$, driven by a dynamical modulation with initial frequency $\Omega_0 = \Omega_{20}^c$ and changing rate ranging from 0 kHz/ns to 75 kHz/ns. It is observed that the magnon state, characterized by center of occupation defined as $\langle\omega\rangle = \omega_0 + \langle n\rangle^2\omega'$ with $\langle n\rangle = \sum_n n\rho_n$, is driven unidirectionally from the initial state $\omega_{20}$ towards $\omega_{28}$ within the time interval of 950 ns for $\kappa/2\pi = 25$ kHz/ns. For larger or smaller $\kappa$, frequency shift becomes less sustainable due to increased instabilities or reduced efficiency, respectively.

It is important to note that there is an order-of-magnitude disparity between the driving frequency $\Omega$ and the resonant frequency $\langle\omega\rangle$, as depicted in Fig. 3b, indicating an overall amplification of frequency shift, or a leverage effect, enabling the realization of a substantial frequency shift on a target by employing a source with a much smaller frequency change. We further define the leverage ratio $\Delta\langle\omega\rangle/\Delta\Omega$ as the ratio of frequency change during the time interval from $t_1 = 50$ ns to $t_2 = 1000$ ns with $\Delta\langle\omega\rangle = \langle\omega\rangle_{t_1} - \langle\omega\rangle_{t_2}$ and $\Delta\Omega = \Omega_{t_1} - \Omega_{t_2}$. As shown in Fig. 3c, the leverage ratio reaches maximum around $\kappa/2\pi = 20$ kHz/ns, comparable to the average hopping velocity of a particle hopping between neighbor sites $(\Omega_n^c - \Omega_{n-1}^c)/2T_B$. Generally speaking, the optimal $\kappa$ should match the natural transition rate

between adjacent frequency modes to ensure smooth and stable mode-to-mode transitions. The numerical results for $\kappa/2\pi = 25$ kHz/ns produced by both TB and MS methods are shown in Fig. 3d, where a symmetric frequency shift in a reversed direction is achieved by letting $\kappa/2\pi = -25$ kHz/ns. Except for the state that is driven unidirectionally, there exists another state that remains almost unchanged around the initial state $\omega_{20}$ as a consequence of the inhomogeneous onsite potential, which can be eliminated by preparing the initial magnon state with a finite momentum in the synthetic frequency dimension (see Methods). This study investigates a ring resonator with a geometry on the order of hundred nanometers, which leads to an optimized changing rate around kHz/ns, too high for realistic application. We highlight that the required changing rate can be further reduced by employing ring resonators with larger geometry, where the spacing between energy levels would be decreased, and the leverage ratio can be further amplified.

## Discussion

One feature of spin waves is the group velocity dispersion which is naturally present even when the dipolar effect is excluded, bringing a vague boundary in the synthetic frequency dimension[55] and leading to nonreciprocal evolution of the magnon state. Another distinct feature of magnetic systems is the rich dynamics of magnetic textures, such as magnetic vortices[56] and magnetic skyrmions[49,57], which support multiple inherent modes below magnon continuum. It is expected to explore a new way to manipulate the excitation of single magnetic texture or quasicrystals such as skyrmion lattices in a hybrid space including realistic spatial dimension (arrays of multiple resonators or magnetic textures) and synthetic frequency dimension. Synthetic pseudospin dimension[34] can be straightforwardly constructed for antiferromagnetic or ferrimagnetic systems where the polarization degree of freedom can be harnessed[10,58]. The synthetic dimension of a single resonator can be further expanded by incorporating thickness-dependent modes, where additional quantization along the thickness direction provides an accessible new degree of freedom. Beyond these potential extensions, it is also important to consider the effects of nonlinearity in practical applications. While magnons experience Gilbert damping losses, our approach operates in the linear spin-wave regime, avoiding the high-energy dissipation typically associated with power-intensive magnetic devices. Although nonlinear effects could enable additional functionalities through processes including multi-magnon scattering[14,22] (analogous to electron-electron interactions in solid state systems), nonlinear frequency shift[59], self-phase modulation[60] and bistability[61], our focus on the linear spin-wave regime ensures reliable and energy efficient operation without requiring high-power thresholds for frequency modulation.

One of the intriguing topics is the effective gauge field[62,63] that magnons hop between spatially separated lattices as well as non-reciprocal hopping induced by intrinsic interactions such as Dzyaloshinskii-Moriya interaction[6]. It needs to be strengthened that the dynamical manipulation is only required during the manipulation of magnon state. When the dynamical modulation is turned off, the effective hopping rate $g$ instantly goes to zero and the magnon state stops to evolve, i.e., stays where it is rather than returning back to the initial state. We anticipate that our work will stimulate further research on the design of magnonic logic gates and other information processing units based on frequency modulation[64,65], where binary information is encoded into different magnon occupation in the spectrum rather than in amplitude or phase.

In conclusion, we explore the manipulation of magnon states in a ring resonator within the frequency domain by leveraging the concept of synthetic frequency dimensions. By deriving an effective tight-binding model from the original micromagnetic model, we predict the evolution of magnon states, including Bloch oscillations and the leverage effect during unidirectional frequency shifts, which are

verified via micromagnetic simulations. Notably, all of these manipulations are valid within the linear regime without the requirement of high-threshold power, which renders them promising for the design of energy-efficient and controllable magnonic devices. Our work potentially opens up a new avenue in magnonics, namely, magnonics in synthetic dimensions.

## Methods

### Tight-binding calculations

The effective tight-binding Hamiltonian is solved by QuTiP[47]. The initial state is set as a Gaussian function in the synthetic frequency dimension, i.e., $\langle n|\psi_0\rangle = \frac{1}{\sqrt{\pi N\sigma}}\exp[-\frac{(n-n_0)^2}{(N\sigma)^2}+ik_0 n]$, with the center of initial distribution $n_0 = 20$, total lattice number $N = 30$, the standard deviation $\sigma = 0.01$ and the initial momentum $k_0 = 0$.

### Micromagnetic simulation using COMSOL

The micromagnetic simulations are performed using the Micromagnetics Module (Time Domain, V2.12) based on COMSOL Multiphysics[48,49]. A magnon state centered at $\omega_{20}$ is injected into the ring resonator by a transverse field pulse applied locally on the magnonic resonator with expression $h_0 \sin(\omega_{20}t)\exp[-(x/\Delta w)^2]$ $\exp-[(t-5/f_{20})/(50/f_{20})]^2$, where $h_0 = 1000$ A/m, $\Delta w = 20$ nm and $\omega_{20} = 2\pi f_{20}$. The dynamical modulation is turned on after $t = 50$ ns and is applied to the sector region of the ring resonator with angle of 20 degrees, so that there is an amplifying factor $360°/20° = 18$ multiplied on the modulation strength $H_m$ for the micromagnetic simulations.

## Data availability

All data needed to evaluate the conclusions in the paper are present in the paper, the Supplementary Information and the Source Data file. Source data are provided with this paper.

## Code availability

The code to solve tight-binding Hamiltonian can be found at: https://github.com/lampiaXu/Frequency-modulation-on-magnons-in-synthetic-dimensions. The Micromagnetics Module developed for COMSOL can be downloaded at https://cn.comsol.com/community/exchange/883/.

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

## Acknowledgements

This work was supported by the Innovation Program for Quantum Science and Technology (Grant No. 2024ZD0300103 to W.Y.), the National Key Research Program of China (Grant No. 2022YFA1403300 to W.Y. and 2022YFA1404204 to Y.C.), the National Natural Science Foundation of China (Grant No. 12204107 to W.Y. and 12274086 to Y.C.), and the Shanghai Science and Technology Committee (Grant No. 21JC1406200 to W.Y.). The authors thank Luqi Yuan for fruitful discussions.

## Author contributions

W.Y. conceived the project. M.X. performed theoretical derivation and tight-binding calculation. C.H. and W.Y. performed micromagnetic simulations. M.X. and W.Y. wrote the manuscript with the help of all the coauthors. All authors contributed to the scientific discussion and commented on the manuscript.

## Competing interests

The authors declare no competing interests.
