## [Transparent Peer Review file · Nature Communications]

Frequency modulation on magnons in synthetic dimensions

Corresponding Author: Dr Weichao Yu

Version 0:

Reviewer comments:

Reviewer #1

(Remarks to the Author)

The work by Xu et al presents the theoretical study of magnon manipulation in the synthetic dimension using frequency modulation. The concept of synthetic dimension has gained increasing interests recently and been adopted in other physical platforms such as photonics, but its application in magnonic platforms has been limited. This work shows a novel approach to introduce synthetic dimension using frequency modulation on a ring-shaped magnonic resonators, and demonstrated the intriguing phenomenon enabled by this approach, such as Bloch oscillation and unidirectional frequency shift. The manuscript is well organized, the model is described with sufficient details, and the analysis is supported by the data obtained using both analytical model and micromagnetic simulation. In general, this work is of high quality so I recommend its acceptance. But the authors should address the following issues first to improve the readability of the manuscript, particularly to the general audience.

1. The description in Line 135 – Line 139 is not very clear. The authors should provide more explanations.
2. Although the SM explains the long-range hopping clearly, the related discussion in Lines 143 – 146 of the main text is not very clear, particularly because a proper definition of the “evolving interval” (first shown in Line 135) is missing. Lines 143—146 is also misleading, suggesting that the long-range hopping (for example, to the next-nearest neighbor) will be enabled by the same driving frequency that causes the hopping to the nearest neighbor, which is not the case according to the discussion in the SM.
3. Line 160 and Fig. 3(a), what determines the optimal κ value to obtain a steady final state?
4. In Lines 177—179, the authors suggest that larger geometries will reduce the parameters to a more realistic range, but their actual demonstration used a small ring resonator ($R=575\text{nm}$), is it limited by the micromagnetic simulation? Also a 2D structure is used in their simulation/modeling. The authors should comment about the effect of the thickness.
5. Line 129, the authors should explain the features of the magnon state evolution with more details. The phrase “widening and shrinking” is a little confusing.
6. The authors should proofread the manuscript for grammatical issues and other minor issues. For example: Line 155, Figure.3(a) should be Figure 3(a). Line 74, “It’s” should be “it is”. Line 144, a blank space is missing before “This”. Fig. 3(c) is not referred to in the main text. Fig. 2(a) should be explained in more detail. All the variables used in the equations and texts should be explained, such as A , L , n which are missing in the current manuscript.

Reviewer #2

(Remarks to the Author)

This paper by Xu et al. introduces the concept of synthetic frequency dimension into magnonics and predicts several intriguing phenomena, including Bloch oscillations and leverage effect during unidirectional frequency shifts. All are verified by micromagnetic simulations. The concept is interesting and the results are convincing. More importantly, to my knowledge, this idea can be verified experimentally with current technology. I am happy to recommend it for publication in Nature Communications after minor revisions.

1. The authors claim that one of the advantages of this concept is that it could operate in the linear region. Could the author also discuss the influence of nonlinearity on this phenomenon?
2. As mentioned in the manuscript, the frequency spacing is unequal. During the oscillation, how can the system maintain the energy conservation?
3. In the theoretical mode, the demagnetizing field is not considered. However, for the FVSW geometry (the external field is

normal to the surface of the sample), the demagnetizing field is important to calculate a correct internal field. Recently, there have been some papers about the nonlinearity in the FVSWs. In the real experiments, the nonlinear shift should also be considered, which is mainly caused by the demagnetizing field. [Sci. Adv. 9, eadg4609 (2023), Nat. Commun. 15, 7577 (2024), Sci. Rep. 12, 7246 (2022).] Could the author compare the simulations with and without demagnetizing field in Mumax3 ?

4. If I am right, the key point of the synthetic frequency concept is the discrete dispersion curve. From the experimental realization point of view, a high-quality ring resonator with a VCMA excitation region is difficult to fabricate. However, the dispersion curve of a simple rectangular Py (1um wide and 5 um long) strip is also discrete. Could the authors comment on this geometry?

Reviewer #3

(Remarks to the Author)

This is a very interesting paper that describes a new formalism for the understanding of magnon dynamics in nanostructures, borrowing techniques from optics. Despite the potential significance of this formalism, I am not convinced that the paper is of interest to a wider audience as it would be expected for a Nat. Comm. paper. Likely, a more specialized physics journal would appreciate this work.

The work has technical inaccuracies, deficiencies or missing links to other works. The paper states that magnonic systems are free from Joule heating – this is true, but magnons also dissipate heat (even if is not called Joule heating) and in most cases magnonic systems are far more lossy than an electronic system.

To my understanding the scheme is based on Floquet theory – a very closely related scheme (Momeni, Ali, and Romain Fleury. "Electromagnetic wave-based extreme deep learning with nonlinear time-Floquet entanglement." Nature Communications 13, no. 1 (2022): 2651.) is not cited. The approach developed by the authors is related to Hamiltonian theory (Krivosik, Pavol, and Carl E. Patton. "Hamiltonian formulation of nonlinear spin-wave dynamics: Theory and applications." Physical Review B—Condensed Matter and Materials Physics 82.18 (2010): 184428.) , which is not mentioned.

Most importantly, the authors use a Comsol Multiphysics-based, frequency domain simulation to validate their approach – this is not a tool that is widely used and trusted by the magnonic community. Simulations by mumax (Vansteenkiste, Arne, Jonathan Leliaert, Mykola Dvornik, Mathias Helsen, Felipe Garcia-Sanchez, and Bartel Van Waeyenberge. "The design and verification of MuMax3." AIP advances 4, no. 10 (2014).) or OOMMF would be a far more convincing demonstration of the approach's validity.

The paper states that 'Demagnetizing field is not considered here for simplicity, which is valid for high-frequency excitations where exchange interaction dominates.' It is almost always a very crude approximation to ignore demagnetization field. To justify such approximation the authors should have done a very careful comparison to mumax-type tools, and this is not done in this paper.

I am also unsure if the consequences of the theory (finding Bloch oscillations or Floquet-mixing) add up to high-impact paper. I do not see even a strawman or a suggestion of a groundbreaking experiment or device application. At the beginning the authors promise that the synthetic dimensions will open up a new degree of freedom to work with in magnonics – but I do not see any groundbreaking new effects, devices that come out from this theory.

Version 1:

Reviewer comments:

Reviewer #2

(Remarks to the Author)

The authors answered my questions satisfactorily and improved the manuscript accordingly, especially the authors checked this idea in a rectangular Py, which is easy to realize in experiments. I believe that this work will soon be confirmed in experiments. I recommend this revised manuscript for publication in Nature Communications.

Reviewer #3

(Remarks to the Author)

The authors did a superb job in addressing my concerns about the first version of the manuscript. The inclusion of Mumax simulations in the supplementary material adds a lot of value to the paper. I am also happy to see the relations to the Hamiltonian and Floquet theory clarified.

I still believe that the results are interesting only to a limited (narrow) audience. Nevertheless, there are no issues with the quality and the clarity of the paper now.

Point-by-point Responses to Reviewer's Comments on Manuscript NCOMMS-24-57816

*All changes in manuscript are highlighted in magenta.

Reply to comments from the Reviewer #1

The work by Xu et al presents the theoretical study of magnon manipulation in the synthetic dimension using frequency modulation. The concept of synthetic dimension has gained increasing interests recently and been adopted in other physical platforms such as photonics, but its application in magnonic platforms has been limited. This work shows a novel approach to introduce synthetic dimension using frequency modulation on a ring-shaped magnonic resonators, and demonstrated the intriguing phenomenon enabled by this approach, such as Bloch oscillation and unidirectional frequency shift. The manuscript is well organized, the model is described with sufficient details, and the analysis is supported by the data obtained using both analytical model and micromagnetic simulation. In general, this work is of high quality so I recommend its acceptance. But the authors should address the following issues first to improve the readability of the manuscript, particularly to the general audience.

Reply: We appreciate the reviewer's positive assessment of our work and his/her constructive suggestions. We have addressed each point as follows.

1. The description in Line 135 – Line 139 is not very clear. The authors should provide more explanations.

Reply: Thanks for pointing this out. We have improved the writing in Lines 135-139 and provide more explanations to clarify the distinct behavior of magnonic systems compared to conventional systems:

“Due to the unequal level spacing inherent in the magnonic system, when the magnon state undergoes transitions between neighboring sites, it experiences multiple hopping

processes with different effective electric fields (Δ_{18} , Δ_{19} , Δ_{21} , Δ_{22} , etc.). These effective fields are larger than Δ_{20} (the one at the initial state ω_{20}). Consequently, when the modulation strength H_m increases, the magnon state can hop more readily between sites experiencing these larger effective fields, resulting in a decreased Bloch oscillation period T_B . This behavior differs from conventional systems with equally spaced levels, where the Bloch period would remain constant regardless of hopping strength.”

2. *Although the SM explains the long-range hopping clearly, the related discussion in Lines 143 – 146 of the main text is not very clear, particularly because a proper definition of the “evolving interval” (first shown in Line 135) is missing. Lines 143—146 is also misleading, suggesting that the long-range hopping (for example, to the next-nearest neighbor) will be enabled by the same driving frequency that causes the hopping to the nearest neighbor, which is not the case according to the discussion in the SM.*

Reply: We acknowledge the confusion regarding the “evolving interval”, and we have replaced the terminology with “hopping range”, which is defined as the total range of frequency sites that a magnon state can sequentially access through nearest-neighbor hopping during its temporal evolution. Additionally, we have revised lines 143-146 to enhance clarity and ensure consistency with the discussion in the SM:

“It is worth noting that when this hopping range becomes large enough, the assumption $\omega_n - \omega_{n-1} < \Omega < \omega_{n+1} - \omega_n$ no longer holds for all involved modes, and additional hopping processes beyond nearest-neighbor transitions become possible. However, these long-range hoppings would require different driving frequencies than those that enable nearest-neighbor transitions.”

3. *Line 160 and Fig. 3(a), what determines the optimal κ value to obtain a steady final state?*

Reply: We thank the reviewer for this important question. The parameter κ represents the rate at which the driving frequency changes over time, and its optimal value is

crucial for achieving stable and efficient frequency shifts of magnon states. The optimal κ value is determined by balancing two competing factors: the speed of frequency shift and the stability of the process. From our analysis, we find that $\kappa/2\pi \approx 25$ kHz/ns provides the best performance, which is close to the theoretical value of $(\Omega_n^c - \Omega_{n-1}^c)/2T_B \approx 23.35$ kHz/ns. This rate allows for efficient mode-to-mode transitions while maintaining system stability.

A faster changing rate could theoretically lead to quicker frequency shifts but would make the system more susceptible to instabilities and less tolerant to deviations from ideal conditions. Conversely, a slower rate would provide better stability but at the cost of operational efficiency. The optimal κ value we identified represents a sweet spot that maximizes the leverage effect (as shown in Fig. 3(c)) while ensuring reliable operation. We have added two sentences to intuitively clarify this point:

“For larger or smaller κ , frequency shift becomes less sustainable due to increased instabilities or reduced efficiency, respectively.”

“Generally speaking, the optimal κ should match the natural transition rate between adjacent frequency modes to ensure smooth and stable mode-to-mode transitions.”

4. In Lines 177—179, the authors suggest that larger geometries will reduce the parameters to a more realistic range, but their actual demonstration used a small ring resonator ($R=575\text{nm}$), is it limited by the micromagnetic simulation? Also a 2D structure is used in their simulation/modeling. The authors should comment about the effect of the thickness.

Reply: The selection of geometry, specifically the small ring size and 2D structure, is not constrained by micromagnetic simulation. Instead, it is a choice to justify the elimination of the demagnetizing field, allowing the exchange interaction to dominate in this scenario, so that the numerical demonstration of the concept can be straightforward and clear.

The effect of thickness warrants discussion from two key perspectives:

- (1) Regarding the additional synthetic dimension, the thickness direction introduces another degree of freedom for spin wave excitations, effectively unlocking an additional dimension in the synthetic lattice. Magnon states in a ring resonator

would need to be characterized by two mode numbers (m, n) , where n represents the azimuthal mode number and m represents the thickness mode number. This presents an interesting avenue for future research. We have added the following to the discussion section:

“The synthetic dimension of a single resonator can be further expanded by incorporating thickness-dependent modes, where additional quantization along the thickness direction provides an accessible new degree of freedom.”

(2) We initially excluded demagnetizing fields in order to clearly demonstrate the underlying principles. However, for more realistic predictions, considering the effects of demagnetizing fields becomes crucial. As suggested by multiple reviewers, we have now incorporated demagnetizing field effects into new micromagnetic simulations using MuMax3, as demonstrated in Fig.R1 below. The discussion below has been added as a new section into the supplementary information:

Figure R1. Micromagnetic simulation of Bloch oscillation performed using MuMax3. (a) In the absence of demagnetizing field, all the parameters are consistent with Fig.2(c) in the main text. (b) The demagnetizing field is present with saturation magnetization $M_s = 0.194 \times 10^6$ A/m and external field $H = 0.159 \times 10^5$ A/m. (c) The demagnetizing field is present with saturation magnetization $M_s = 0.194 \times 10^6$ A/m and external field $H = 0.339 \times 10^5$ A/m. In all cases, the frequency of dynamic modulation is chosen as $\Omega_{20}^c = (\omega_{21} - \omega_{19})/2$.

“Figure R1 shows the same conditions as Fig.2(c) without the demagnetizing field, demonstrating perfect agreement with the COMSOL simulation. For cases including the demagnetizing field, we analyze a ring resonator with finite thickness $d = 10$ nm and saturation magnetization $M_s = 0.194 \times 10^6$ A/m. We examine two scenarios:

(i) With an external field $H = 0.159 \times 10^5$ A/m, comparable to the internal demagnetizing field, Bloch oscillation occurs but with an indeterminate period, as shown in Fig.R1(b). This irregularity arises because in this demagnetized-field-dominated regime, the working frequency approaches the spin-wave gap, resulting in more closely spaced energy levels compared to the exchange-dominated regime.

(ii) With a stronger external field $H = 0.339 \times 10^5$ A/m, exceeding the demagnetizing field (though its contribution remains significant), the exchange interaction dominates and produces stable Bloch oscillation, as shown in Fig.R1(c). However, the higher working frequency induced by the stronger external field leads to faster dynamics and increased damping over the same time period. Consequently, the spin-wave amplitude decays more rapidly than in Fig.R1(a).”

We are grateful for the reviewer’s insightful comments to bring these important considerations to our attention and the discussion on the effect of demagnetizing field has been added in the supplementary information.

5. Line 129, the authors should explain the features of the magnon state evolution with more details. The phrase “widening and shrinking” is a little confusing.

Reply: We have revised the description of magnon state evolution, replacing the original sentence with more precise description as:

“The magnon state initially localized at ω_{20} begins to spread across neighboring frequency modes, populating states from ω_{19} to ω_{21} . This spreading and subsequent reconcentration occurs periodically with the Bloch period T_B .”

6. The authors should proofread the manuscript for grammatical issues and other minor issues. For example: Line 155, Figure.3(a) should be Figure 3(a). Line 74, “It’s” should be “it is”. Line 144, a blank space is missing before “This”. Fig. 3(c) is not referred to in the main text. Fig. 2(a) should be explained in more detail. All the

variables used in the equations and texts should be explained, such as A , L , n which are missing in the current manuscript.

Reply: We thank the reviewer for careful reading. We have thoroughly proofread the manuscript and corrected all the issues. Figure 2(a) as well as its inset has been explained in more detail:

“The inset of Fig.2(a) specifically demonstrates how the critical modulation frequency Ω_{20}^c creates an energy configuration where the dressed state (under modulation) and the original magnon state at ω_{20} form equally-spaced energy levels, which is essential for achieving coherent Bloch oscillation.”

Reply to comments from the Reviewer #2

This paper by Xu et al. introduces the concept of synthetic frequency dimension into magnonics and predicts several intriguing phenomena, including Bloch oscillations and leverage effect during unidirectional frequency shifts. All are verified by micromagnetic simulations. The concept is interesting and the results are convincing. More importantly, to my knowledge, this idea can be verified experimentally with current technology. I am happy to recommend it for publication in Nature Communications after minor revisions.

Reply: We sincerely thank the reviewer for his/her positive assessment of our work and the recommendation for publication. We have carefully considered the following feedback and revised the manuscript accordingly.

1. The authors claim that one of the advantages of this concept is that it could operate in the linear region. Could the author also discuss the influence of nonlinearity on this phenomenon?

Reply: We thank the reviewer for this important question about nonlinear effects. Indeed, one key advantage of our approach is its operation in the linear spin-wave regime, which avoids complications associated with nonlinear effects. We expect that moving into the nonlinear regime would introduce both additional functionalities and complexities:

(1) There will be amplitude-dependent frequency shifts for the resonant modes, which could distort the levels in the effective tight-binding model. However, the concept is still available as long as the levels are determinate.

(2) The mode coupling through multi-magnon processes could potentially enable new functionalities, as it provides additional channels for magnon-magnon interactions, though it may also lead to chaotic dynamics and reduced controllability of the frequency transitions. This process is analogous to electron-electron interactions in solid state systems, where many-body effects can give rise to rich emergent phenomena while modifying the simple tight-binding picture.

While nonlinear effects could enable new functionalities, our current focus on the linear regime ensures reliable and energy-efficient operation without requiring high-power thresholds. We have modified the manuscript to include this detailed discussion on the influence of nonlinearity on the phenomenon. We have added the discussion on nonlinear effects in the revised manuscript:

“While nonlinear effects could enable additional functionalities through processes including multi-magnon scattering (analogous to electron-electron interactions in solid state systems), nonlinear frequency shift, self-phase modulation and bistability, our focus on the linear regime ensures reliable and energy-efficient operation.”

2. As mentioned in the manuscript, the frequency spacing is unequal. During the oscillation, how can the system maintain the energy conservation?

Reply: This is an insightful question. In the absence of Gilbert damping and dynamic modulation, the energy of the system increases quadratically with the mode number n . Therefore, the total energy of the isolated magnonic system cannot remain constant during either Bloch oscillation or unidirectional frequency shift, as shown in Fig.R2, where the expected energy of a magnon is defined as $E(t) = \hbar \sum_n \rho_n(n, t) \omega_n$.

Figure R2. The evolution of energy during (a) Bloch oscillation (Fig. 2(b) in the main text) and (b) unidirectional frequency shift (Fig.3(d) in the main text).

In fact, the dynamical modulation serves as an energy mediator, injecting and extracting energy from the magnetic system through alternating positive and negative work. This ensures that the total energy of the complete system (comprising both the magnetic

system and the modulation device) remains conserved. We have included an explanation regarding to energy conservation:

“The dynamical modulation provides the necessary energy compensation through the coupling term g in Eq.(7), enabling coherent transitions between adjacent modes with unequal frequency spacing while maintaining overall energy conservation through balanced energy exchange between the magnetic system and modulation device.”

3. In the theoretical model, the demagnetizing field is not considered. However, for the FVSW geometry (the external field is normal to the surface of the sample), the demagnetizing field is important to calculate a correct internal field. Recently, there have been some papers about the nonlinearity in the FVSWs. In the real experiments, the nonlinear shift should also be considered, which is mainly caused by the demagnetizing field. [Sci. Adv. 9, eadg4609 (2023), Nat. Commun. 15, 7577 (2024), Sci. Rep. 12, 7246 (2022).] Could the author compare the simulations with and without demagnetizing field in MuMax3 ?

Reply: We appreciate the reviewer for highlighting these relevant works on novel effects when transitioning into the nonlinear regime. We have appropriately cited these references in our discussion of nonlinear effects. These effects become apparent in the presence of demagnetizing field. We followed the reviewer’s suggestion and conducted micromagnetic simulations using MuMax3, both with and without the demagnetizing field, which have been added as a new section into the supplementary information:

“Figure R1 shows the same conditions as Fig.2(c) without the demagnetizing field, demonstrating perfect agreement with the COMSOL simulation. For cases including the demagnetizing field, we analyze a ring resonator with finite thickness $d = 10$ nm and saturation magnetization $M_s = 0.194 \times 10^6$ A/m. We examine two scenarios:

(i) With an external field $H = 0.159 \times 10^5$ A/m, comparable to the internal demagnetizing field, Bloch oscillation occurs but with an indeterminate period, as shown in Fig.R1(b). This irregularity arises because in this demagnetized-field-dominated regime, the working frequency approaches the spin-wave gap, resulting in more closely spaced energy levels compared to the exchange-dominated regime.

(ii) With a stronger external field $H = 0.339 \times 10^5$ A/m, exceeding the demagnetizing field (though its contribution remains significant), the exchange interaction dominates

and produces stable Bloch oscillation, as shown in Fig.R1(c). However, the higher working frequency induced by the stronger external field leads to faster dynamics and increased damping over the same time period. Consequently, the spin-wave amplitude decays more rapidly than in Fig.R1(a).”

Figure R1. Micromagnetic simulation of Bloch oscillation performed using MuMax3. (a) In the absence of demagnetizing field, all the parameters are consistent with Fig.2(c) in the main text. (b) The demagnetizing field is present with saturation magnetization $M_s = 0.194 \times 10^6$ A/m and external field $H = 0.159 \times 10^5$ A/m. (c) The demagnetizing field is present with saturation magnetization $M_s = 0.194 \times 10^6$ A/m and external field $H = 0.339 \times 10^5$ A/m. In all cases, the frequency of dynamic modulation is chosen as $\Omega_{20}^c = (\omega_{21} - \omega_{19})/2$.

4. If I am right, the key point of the synthetic frequency concept is the discrete dispersion curve. From the experimental realization point of view, a high-quality ring resonator with a VCMA excitation region is difficult to fabricate. However, the dispersion curve of a simple rectangular Py (1um wide and 5 um long) strip is also discrete. Could the authors comment on this geometry?

Reply: We appreciate the reviewer's practical suggestion. A rectangular stripe indeed offers a simpler structure that is easier to fabricate. The reviewer correctly identifies the key concept: the synthetic frequency dimension approach can be applied to any wave-based system that preserves discrete eigenstates. To validate this suggestion, we performed micromagnetic simulations using MuMax3 on an in-plane magnetized stripe

in the presence of demagnetizing field. We maintained consistency with the parameters used in the main text for qualitative comparison.

While the concept of synthetic frequency dimension is valid across all regimes, achieving coherent and stable frequency transitions requires sufficient separation between energy levels that exceeds the linewidth induced by intrinsic damping. We investigated stripes with different geometries, as shown in Fig.R3. The geometry in Fig.R3(a), as proposed by the reviewer, has a relatively large width that allows standing wave modes in the width direction to overlap with longitudinal modes, evidenced by multiple sidebands in the spectrum. The geometry in Fig.R3(b) is also suboptimal. Although the standing wave modes in the width direction are significantly elevated, the spacing between longitudinal modes remains too narrow. Consequently, we selected the geometry shown in Fig.R3(c) to demonstrate Bloch oscillation, as presented in Fig.R4.

Figure R3. Resonant state of an in-plane magnetized stripe calculated using MuMax3 for different geometries with (a) 5000nm × 1000nm, (b) 5000nm × 100nm and (c) 3000nm × 100nm. The horizontal axis n is equivalent to wave number k for a closed system.

Figure R4. Bloch oscillation for an in-plane magnetized stripe with geometry 3000nm × 100nm. The frequency of dynamic modulation is chosen as $\Omega_{20}^c = (\omega_{21} - \omega_{19})/2$.

Reply to comments from the Reviewer #3

This is a very interesting paper that describes a new formalism for the understanding of magnon dynamics in nanostructures, borrowing techniques from optics. Despite the potential significance of this formalism, I am not convinced that the paper is of interest to a wider audience as it would be expected for a Nat. Comm. paper. Likely, a more specialized physics journal would appreciate this work.

Reply: We sincerely thank the reviewer for recognizing the potential significance of our work and appreciate the reviewer's critical assessment. Below, we have addressed the concerns, technical inaccuracies and deficiencies according to the reviewer's feedback. Furthermore, we would like to emphasize several key aspects that demonstrate the broad significance of our work.

The work has technical inaccuracies, deficiencies or missing links to other works. The paper states that magnonic systems are free from Joule heating – this is true, but magnons also dissipate heat (even if is not called Joule heating) and in most cases magnonic systems are far more lossy than an electronic system.

Reply: We agree with the reviewer that magnons do dissipate heat in the form of Gilbert damping. Nevertheless, in certain magnetic dielectrics with low damping properties, such as YIG (yttrium iron garnet), magnons can propagate over centimeter-scale distances with minimal dissipation. This remarkable property serves as one of the fundamental principles underlying the field of magnon spintronics (Chumak et al., Nature Physics 11, 453-461 (2015)). By stating “*in most cases magnonic systems are far more lossy than an electronic system*”, we believe the reviewer is referring to scenarios where power-intensive microwave devices employ magnetic materials for isolation purposes, or where magnetic cores handle substantial power loads in high-frequency electronics. In such applications, the combination of highly damped magnetic materials operating in high-power regimes inevitably leads to substantial energy losses. Notably, our work specifically addresses this challenge by achieving frequency modulation in the GHz regime while operating in the linear spin-wave

regime, thereby eliminating the need for high-power injection and, consequently, energy losses. We have added a paragraph in the discussion section to emphasize that our approach operates in the linear spin-wave regime, which offers significant advantages in terms of energy efficiency:

“While magnons experience Gilbert damping losses, our approach operates in the linear spin-wave regime, avoiding the high energy dissipation typically associated with power-intensive magnetic devices.”

*To my understanding the scheme is based on Floquet theory – a very closely related scheme (Momeni, Ali, and Romain Fleury. "Electromagnetic wave-based extreme deep learning with nonlinear time-Floquet entanglement." *Nature Communications* 13, no. 1 (2022): 2651.) is not cited. The approach developed by the authors is related to Hamiltonian theory (Krivosik, Pavol, and Carl E. Patton. "Hamiltonian formulation of nonlinear spin-wave dynamics: Theory and applications." *Physical Review B — Condensed Matter and Materials Physics* 82.18 (2010): 184428.) , which is not mentioned.*

Reply: Thanks for pointing out these references. Our work indeed shares foundations with Floquet theory, particularly in our utilization of periodic time-dependent modulation and gauge transformation to derive the effective Hamiltonian. We have properly cited these references in the revised manuscript. While acknowledging the connections, we would like to highlight the key distinctions:

- (1) Momeni and Fleury applied Floquet theory in electromagnetic wave systems, demonstrating how periodic modulation can generate robust, tunable nonlinear entanglement between frequency-diverse signal inputs. Our work, however, explores frequency modulation of spin waves in synthetic frequency dimensions, rather than in real space. This fundamental difference leads to a crucial distinction: when dynamical modulation (Floquet driving) ceases, the nonlinear signal entanglement in Momeni and Fleury's work disappears, whereas in our system, magnon states maintain their positions rather than reverting to their initial state.
- (2) Krivosik and Patton developed a comprehensive Hamiltonian formalism for nonlinear spin-wave dynamics, extending spin-wave interactions to fourth-order terms and transforming torque equations into scalar Hamiltonian form. While both approaches employ Hamiltonian formalism, they differ fundamentally in concept

and implementation. Their work utilizes conventional Hamiltonian formalism, where canonical variables represent temporal and spatial magnetization vector variations. In contrast, our approach derives an effective tight-binding Hamiltonian from the Landau-Lifshitz-Gilbert equation within the synthetic frequency dimension framework, where creation and annihilation operators correspond to magnon occupation in frequency (mode) states. These approaches diverge significantly in their theoretical foundations and physical interpretations.

In a brief summary, while our work shares theoretical foundations with these seminal references, it presents a novel approach focused on magnon manipulation in the linear spin-wave regime through the concept of synthetic frequency dimensions, establishing a new paradigm for manipulating magnons in frequency dimension.

Most importantly, the authors use a Comsol Multiphysics-based, frequency domain simulation to validate their approach – this is not a tool that is widely used and trusted by the magnonic community. Simulations by mumax (Vansteenkiste, Arne, Jonathan Leliaert, Mykola Dvornik, Mathias Helsen, Felipe Garcia-Sanchez, and Bartel Van Waeyenberge. "The design and verification of MuMax3." AIP advances 4, no. 10 (2014).) or OOMMF would be a far more convincing demonstration of the approach's validity.

Reply: We validated our theory using the COMSOL-based micromagnetic module (*time domain*), which directly solves the original LLG equation in contrast to the micromagnetic module (*frequency domain*) which employs a linearized version of LLG equation. The misunderstanding may come from our previously cited reference [47] (Zhang et al. A frequency-domain micromagnetic simulation module based on COMSOL Multiphysics, AIP Advances, 2023) which primarily explores frequency-domain simulation capabilities built upon an established time-domain framework. In the revised manuscript, we have cited two more references (Lan et al. Physical Review X, 2015; Lan et al. Nature Communications, 2017), which are conducted using the time-domain module and part of which were later verified by experiments. On the other hand, we acknowledge the reviewer's concern on the validity of the numerical tool. We have followed the suggestion to perform micromagnetic simulation using MuMax3, which are consistent with results produced by COMSOL (to be demonstrated later).

The paper states that 'Demagnetizing field is not considered here for simplicity, which is valid for high-frequency excitations where exchange interaction dominates.' It is almost always a very crude approximation to ignore demagnetization field. To justify such approximation the authors should have done a very careful comparison to mumax-type tools, and this is not done in this paper.

Reply: We appreciate the reviewer's concern regarding the omission of the demagnetizing field. While incorporating demagnetization effects would indeed provide a more comprehensive physical description, our simplified approach is well-justified for several reasons:

- (1) Our investigation primarily focuses on high-frequency exchange spin waves in a nanoscale ring resonator, where exchange interactions dominate over dipolar effects. At these characteristic length scales and frequencies, the exchange contribution to the spin wave dispersion significantly exceeds the dipolar contribution.
- (2) The exclusion of the demagnetizing field enables us to demonstrate the concept of synthetic frequency dimension with greater clarity, allowing the effective tight-binding Hamiltonian to be expressed in an elegant and mathematically tractable form.
- (3) Importantly, the inclusion of the demagnetizing field does not compromise the validity of our theoretical framework. While the increased complexity of spin-wave dispersion in the presence of demagnetizing fields necessitates modified driving strategies, both our fundamental conclusions and methodological approach remain robust.

To address the reviewer's concerns quantitatively, we conducted comprehensive micromagnetic simulations using MuMax3, comparing scenarios both with and without the demagnetizing field. As shown in Fig.R1 below, our findings demonstrate that: (i) In the absence of demagnetizing field, the simulation results from COMSOL and MuMax3 show excellent agreement, validating our numerical approach. (ii) More significantly, when including the demagnetizing field, our principal conclusions remain valid. For instance, we successfully demonstrated that Bloch oscillation can be achieved through appropriate adjustment of the dynamic modulation frequency. The discussion below has been added as a new section into the supplementary information:

“Figure R1 shows the same conditions as Fig.2(c) without the demagnetizing field, demonstrating perfect agreement with the COMSOL simulation. For cases including the demagnetizing field, we analyze a ring resonator with finite thickness $d = 10$ nm and saturation magnetization $M_s = 0.194 \times 10^6$ A/m. We examine two scenarios:

(i) With an external field $H = 0.159 \times 10^5$ A/m, comparable to the internal demagnetizing field, Bloch oscillation occurs but with an indeterminate period, as shown in Fig.R1(b). This irregularity arises because in this demagnetized-field-dominated regime, the working frequency approaches the spin-wave gap, resulting in more closely spaced energy levels compared to the exchange-dominated regime.

(ii) With a stronger external field $H = 0.339 \times 10^5$ A/m, exceeding the demagnetizing field (though its contribution remains significant), the exchange interaction dominates and produces stable Bloch oscillation, as shown in Fig.R1(c). However, the higher working frequency induced by the stronger external field leads to faster dynamics and increased damping over the same time period. Consequently, the spin-wave amplitude decays more rapidly than in Fig.R1(a).”

Figure R1. Micromagnetic simulation of Bloch oscillation performed using MuMax3. (a) In the absence of demagnetizing field, all the parameters are consistent with Fig.2(c) in the main text. (b) The demagnetizing field is present with saturation magnetization $M_s = 0.194 \times 10^6$ A/m and external field $H = 0.159 \times 10^5$ A/m. (c) The demagnetizing field is present with saturation magnetization $M_s = 0.194 \times 10^6$ A/m and external field $H = 0.339 \times 10^5$ A/m. In all cases, the frequency of dynamic modulation is chosen as $\Omega_{20}^c = (\omega_{21} - \omega_{19})/2$.

I am also unsure if the consequences of the theory (finding Bloch oscillations or Floquet-mixing) add up to high-impact paper. I do not see even a strawman or a suggestion of a groundbreaking experiment or device application. At the beginning the authors promise that the synthetic dimensions will open up a new degree of freedom to work with in magnonics – but I do not see any groundbreaking new effects, devices that come out from this theory.

Reply: Thanks for this important feedback. While we respectfully disagree that the work lacks impact, we acknowledge that we could better emphasize its significance and practical implications. Here are several key points that demonstrate the novelty and potential impact of our work:

(1) This work does predict groundbreaking new effects:

- (i) Magnon Bloch oscillation ---- Magnon state (occupation of resonant mode) oscillates in synthetic frequency dimension in the presence of dynamic modulation. This effect is distinct from conventional magnon Bloch oscillation in real space, such as in a spin chain (Kosevich et al., J. Phys.: Condens. Matter 25 246002 (2013)) or in magnonic crystals (Tartakovskaya et al., Low Temp. Phys. 46, 830–835 (2020)). In our approach, the frequency transitions are persistent, i.e., when the modulation is turned off, the magnon state remains at its new frequency rather than returning to its initial state.
- (ii) Unidirectional frequency shift ---- Magnon state (occupation of resonant mode) is unidirectionally shifted under dynamic modulation. This effect is distinct from the well-known nonlinear frequency shift, which occurs when high-amplitude spin waves cause a deviation from their linear dispersion relationship due to strong magnon-magnon interactions, resulting in a frequency-dependent shift that is proportional to the wave intensity. In contrast, our strategy can operate in *linear spin-wave regime*, without necessities of multi-magnon scattering. Importantly, this effect is not directly relevant to Floquet mixing, which creates hybrid states that are superpositions of the original states in the presence of periodic driving. In our approach, the energy levels of magnon states are unperturbed, and the effect of dynamic modulation is to modify the magnon occupation of each resonant state.

(2) This work does provide a strawman device application, i.e., the magnonic ring

resonator, which is a building block of magnonic networks (Wang et al., Phys. Rev. Applied 21, 040503 (2024)) and has wide applications as a frequency-selective filter, a power limiter and an artificial neuron for neuromorphic computing. Our theoretical work takes this simple but functional device as example, and further unlocks its functionality in the linear spin-wave regime. The concept can be further extended to other devices, for example as mentioned by Reviewer #2, a simple nano-stripe supporting discrete standing wave modes can work as a simplest computing unit where frequency modulation can be performed.

In summary, our work introduces the concept of synthetic dimension to magnonics, establishing a theoretical framework that reveals novel phenomena previously unexplored in magnonic systems and distinct from their photonic counterparts. We demonstrate a novel approach for manipulating magnon states in the frequency domain through dynamic modulation, operating within the *linear spin-wave regime*. This strategy opens new possibilities for controlling spin waves without the constraints of traditional multi-magnon scattering methods, potentially advancing the field of magnetic computing and information processing. We thank the reviewer again for his/her thoughtful feedback, which has helped us better explain the significance of our work, and we hope that our revised manuscript now shows the impact needed for publication in Nature Communications.